# Disease activity trajectories for early and established rheumatoid arthritis: Real-world data from a rheumatoid arthritis cohort

**Mohammad Movahedi**[1,2], **Angela Cesta**[1], **Xiuying Li**[1], **Claire Bombardier**[1,2,3]*, **OBRI investigators**[¶]

**1** Toronto General Hospital Research Institute, University Health Network, Toronto, Ontario, Canada,
**2** Institute of Health Policy, Management and Evaluation, University of Toronto, Toronto, Ontario, Canada,
**3** Department of Medicine, University of Toronto, Toronto, Ontario, Canada

¶ Membership of the OBRI investigators is listed in the Acknowledgments.
* claire.bombardier@utoronto.ca

## Abstract

### Objectives

Disease activity status described at fixed time points does not accurately reflect disease course in chronic and relapsing diseases such as rheumatoid arthritis (RA). We described longitudinal disease activity trajectories in early and established RA.

### Methods

Patients with available 28-Joint Disease Activity Score-erythrocyte sedimentation rate (DAS28-ESR) and Clinical Disease Activity Index (CDAI) over two years were included. Using latent growth curve modelling (LCGM), subgroups of patients following distinct patterns were identified.

### Results

1920 patients were included with 34.4% in early RA (< 2 years' disease duration). Three subgroups were identified using DAS28-ESR in early RA: 1) low disease activity to remission (LDA-REM: 19.1%); 2) moderate disease to remission (MD-REM: 54%); 3) high to moderate disease (HD-MD: 26.9%). The HD-MD group had a significantly higher number of comorbidities, biologic and steroid use and lower post-secondary education. Using CDAI, we identified seven subgroups with only 1.9% remission in early RA. In established RA, seven subgroups were identified using either DAS28-ESR or CDAI. Using DAS28-ESR 27.8% with HD showed improvement in disease status (14.2% HD-REM, 10.3% HD-LDA and 3.3% HD-MD) while using CDAI 17.9% showed improvement.

### Conclusion

Disease course was different in early and established RA. Only 14.2% of established RA reached DAS28-ESR remission compared to 73.1% of early RA. Using CDAI only 1.9% of early RA and none of the established RA achieved remission, likely reflecting the impact of

**Data Availability Statement:** Data cannot be shared publicly because participants of this study did not agree for their data to be shared publicly. Data are available from the University Health

Network (UHN) Research Ethics Board (REB) for researchers who meet the criteria for access to confidential data."

**Funding:** The authors received no specific funding for this work.

**Competing interests:** I have read the journal's policy and the authors of this manuscript have the following competing interests: OBRI was funded by peer-reviewed grants from CIHR (Canadian Institute for Health Research), Ontario Ministry of Health and Long-Term Care (MOHLTC), Canadian Arthritis Network (CAN) and unrestricted grants from: Abbvie, Amgen, Aurora, Bristol-Meyers Squibb, Celgene, Gilead, Hospira, Janssen, Lilly, Medexus, Merck, Novartis, Pfizer, Roche, Sandoz, Sanofi, & UCB Acknowledgment: Dr. Bombardier held a Canada Research Chair in Knowledge Transfer for Musculoskeletal Care and a Pfizer Research Chair in Rheumatology Disclosure statement: MM is an employee at OBRI, University Health Network, and has a faculty position (status) at the University of Toronto with no conflict of interest; AC and XL are employees at OBRI reporting no conflict of interest. CB is principal investigator in the OBRI and held a Canada Research Chair in Knowledge Transfer for Musculoskeletal Care.

the patient global assessment on this score. Findings also illustrate the impact of sociodemographic characteristics and early treatment on disease course.

## Introduction

Disease activity profiles vary overtime within and between individual patients with rheumatoid arthritis (RA). Thus, describing disease activity status at fixed time points modelled as continuous or dichotomous variable (e.g. remission (REM) or low disease activity (LDA)) does not reflect the patient's disease course in chronic and relapsing RA. Previous studies have looked at disease trajectories over time mostly using disease activity score-28 (DAS28) in early RA [1] or after biologic treatment initiation [2, 3]. However, disease course may be different in early and established disease. For example, established RA patients are less likely to be biologic naïve, and more likely to be older, have more comorbidities, and use more polypharmacy, all of which would affect disease management in this group of patients. Moreover, considering other disease activity composite measures such as clinical disease activity index (CDAI), which is commonly used in routine clinical practice due to non-reliance on acute phase reactant [4], may reveal different patterns of disease course for patients. These potential differences may have an impact on the treat to target strategy which aims to improve health outcomes of patients with RA.

In this study, we aimed to identify disease trajectories for both patients with early and established RA using the two most common composite measures of diseases activity; DAS28- erythrocyte sedimentation rate (ESR) and CDAI.

## Methods

### Data source

The Ontario Best Practice Research Initiative (OBRI) is a multicenter registry across Ontario, Canada, collecting data from both rheumatologists and patients with RA at enrolment and at follow-up. It incorporates rheumatologist assessments from approximately one-third of the rheumatologists in the province of Ontario. Patients are eligible to participate if they are $\geq 16$ years of age at the time of diagnosis, $\geq 18$ years of age at enrolment, and have a rheumatologist confirmed RA diagnosis. Enrolled patients are interviewed every 6 months by phone and are seen by their rheumatologist as per routine care.

### Data collection

At enrolment, patients are asked about their general medical history and comorbidities, including cardiovascular disease (CVD), RA disease activity and inflammatory markers. Tender and swollen joint counts, data on socio-demographics, smoking status, height, weight as well as any prior and current medications are collected during the rheumatologist enrolment visit or during the patient's interview. Patient-reported outcomes for functional status are also collected.

At follow-up visits, all the aforementioned information is updated. RA medication changes (including discontinuation and reasons for discontinuation) between visits are also captured. Rheumatologists report any incident of comorbidity and re-assess disease activity during every follow-up visit.

For this study, patients enrolled in the OBRI between 1st Jan 2008 and 1st Jan 2020 were included and categorized as early (disease duration since diagnosis < 2 year) or established RA

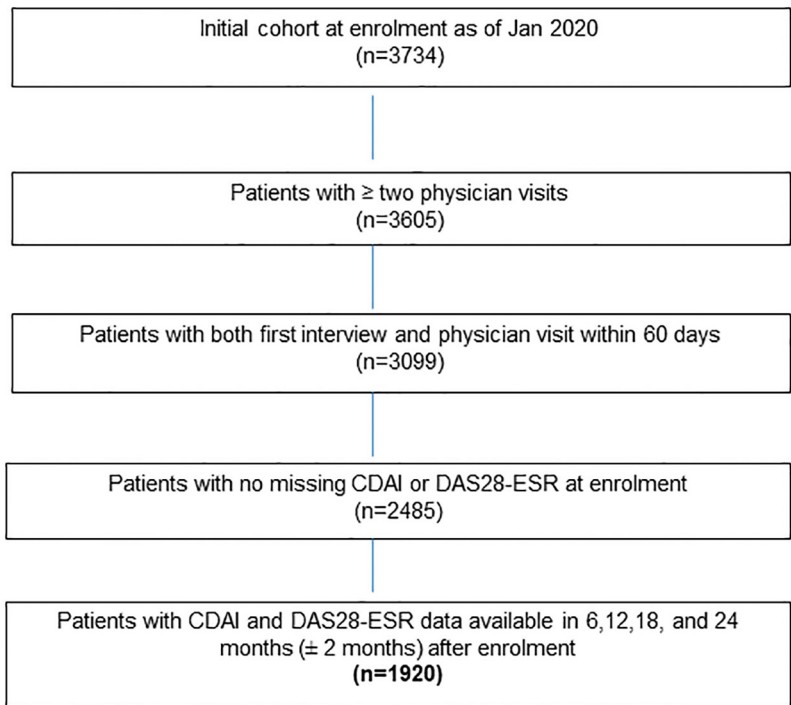

**Fig 1. Cohort flowchart.**

(disease duration since diagnosis $\geq$ 2 year). Patients must also have had at least 2 physician visits and disease activity data (DAS28-ESR and CDAI) available over two years of follow-up (Fig 1).

We defined disease activity based on DAS28-ESR as: Remission (REM): DAS-ESR< = 2.6; LDA: DAS-ESR< = 3.2; Moderate (MD): 3.2<DAS28-ESR< = 5.0; Severe (HD): DAS28-ESR>5.0.

We also defined disease activity based on CDAI as: Remission (REM) CDAI< = 2.8; LDA: CDA< = 10; Moderate (MD): 10<CDAI< = 22; Severe (HD): CDAI>22.

**Ethic statement.**   All sites had ethics approval to enroll patients. All patients signed informed consent as below:

1. Consent was informed

2. Consent was written

3. Consent did not include minors, i.e., participants had to be 18 years of age or older.

    Ethics approval: REB# is 07–0729 AE (University Health Network).

## Data analysis

Baseline demographics and disease parameters were described using means and standard deviation (SD) for continuous variables or counts and proportions for categorical variables. Main comorbidity was defined as the presence of hypertension, cardiovascular disease (CVD), Diabetes mellitus, Lung diseases, cancer, gastrointestinal disease, and depression. Variables in the early and established groups were compared using the t-test or Wilcoxon Rank-Sum test for continuous variables and the chi-square or Fisher's Exact test for categorical variables.

Using latent growth curve modelling (LCGM) and a semi-parametric statistical technique proposed by Nagin [5], subgroups of patients following distinct patterns of DAS28-ESR or CDAI change over time were identified. To specify the shape of each trajectory, a single quadratic trajectory model was first tested. If the quadratic component of this model was significant, the analysis for the quadratic model for two trajectories was performed. This process was repeated with an increasing number of trajectories until the model of best fit was obtained, as determined by comparing the Bayesian information criterion (BIC) values [6]. A low BIC indicates the best fitting distribution and number of trajectory subgroups to describe the data. Just briefly, fit statistics and model selection was based on "log Bayes factor which is calculated as:

$$2 \times [((\text{BIC for current model}) - (\text{BIC for previous model})]".$$

If log Bayes factor is a negative value, we stop and select the previous model. If log Bayes factor is a positive value, we move to the next model by adding another group [5, 7] (Table A1 in S1 Appendix as example). Subjects are then assigned to the group they most likely belong, the criterion (e.g. based on that group being estimated to have the highest posterior probability of the subject being allocated to it) that is used to make this allocation.

Primary analysis identified trajectory subgroups in patients with early and established RA using DAS28- ESR, separately.

Secondary analysis was conducted to identify disease trajectory subgroups based on CDAI for both early and established RA.

Additionally, we compared sociodemographic, disease and treatment variables between trajectory subgroups identified for DAS28-ESR in patients with early RA, by using one-way ANOVA or Kruskal-Wallis test for continuous variables and the chi-square or Fisher's Exact test for categorical variables. The analysis was carried out in SAS (version 9.4) using the "proc traj" application which used a general quasi-Newton procedure to estimate parameters that maximize the likelihood function [1, 6, 7].

## Results

A total of 1920 patients were included, 660 (34.4%) with early and 1260 (65.6%) with established RA (Table 1). At baseline, patients with early RA were significantly younger (mean 56.6 vs. 58.9 years) had higher DAS28-ESR (mean 4.6 vs. 4.1), CDAI (mean 22.8 vs. 19.4), higher ESR (mean 25.3 vs. 22.2), C-reactive protein (CRP) (mean 14.5 vs. 11.2), and were more likely to use concurrent steroids (23.5% vs. 17.4%). These patients were also less likely to have an erosion (24.0% vs. 58.4%), to be RF-positive (68.9% vs. 74.0%), to use prior biologic disease-modifying antirheumatic drugs (bDMARDs) (9.5% vs. 39.0%), and to start new bDMARDs at enrolment (8.8% vs. 29.9%). There was no significant difference in average number of visits between two groups (mean 13.0 vs. 14.0).

### Disease trajectories in early RA

**DAS28-ESR.** In patients with early RA, three subgroups of patients were identified by LCGM (BIC: -5455.78) (Table A1 in S1 Appendix). Group 1 with Low disease activity (LDA) reached remission rapidly by 6 months and remained in this state at 2 years (LDA-REM: 19.1%). Group 2 with moderate disease (MD) improved to LDA at 6 months and then gradually to REM at 2 years (MD-REM: 54%). Group 3 with high disease (HD) showed slight improvement to moderate disease (MD) state over two years (HD-MD: 26.9%) (Fig 2A and Table 2). Overall, all patients with early RA showed an improvement in their disease activity status over two years of follow-up, with 73.1% reaching remission.

**Table 1. Baseline characteristics of patients with RA.**

| | | Disease onset status at enrolment | | |
|---|---|---|---|---|
| | Total (N = 1,920) | Early RA (< 2 years) (N = 660) | Established RA (≥ 2 years) (N = 1260) | P Value |
| Female (%) | 1506 (78.4) | 487 (73.8) | 1019 (80.9) | < .001 |
| Age, years, Mean ± SD | 58.1 ± 12.6 | 56.6 ± 13.3 | 58.9 ± 12.1 | < .001 |
| Marital status, married (%) | 1340 (69.8) | 471 (71.4) | 869 (69.0) | 0.257 |
| Post-secondary education (%) | 1081 (56.3) | 379 (57.4) | 702 (55.7) | 0.474 |
| Current smoker (%) | 318 (16.6) | 110 (16.7) | 208 (16.5) | 0.960 |
| Disease duration, years, Mean ± SD | 7.9 ± 9.4 | 0.3 ± 0.5 | 11.9 ± 9.5 | < .001 |
| PtGA, Mean ± SD | 4.8 ± 2.8 | 5.1 ± 2.7 | 4.6 ± 2.8 | 0.008 |
| PhGA, Mean ± SD | 4.3 ± 2.5 | 4.7 ± 2.4 | 4.0 ± 2.5 | < .001 |
| 28SJC, Mean ± SD | 5.4 ± 4.9 | 5.8 ± 4.9 | 5.2 ± 4.8 | 0.273 |
| 28TJC, Mean ± SD | 6.1 ± 6.2 | 7.1 ± 6.5 | 5.7 ± 6.0 | 0.005 |
| CDAI, Mean ± SD | 20.5 ± 13.5 | 22.8 ± 13.6 | 19.4 ± 13.4 | 0.003 |
| CDAI LDA/REM (CDAI< = 10) (%) | 517 (26.9) | 130 (19.7) | 387 (30.7) | < .001 |
| CDAI REM (CDAI< = 2.8) (%) | 85 (4.4) | 11 (1.7) | 74 (5.9) | < .001 |
| DAS28-ESR, Mean ± SD | 4.3 ± 1.6 | 4.6 ± 1.5 | 4.1 ± 1.6 | < .001 |
| DAS28-ESR LDA/REM (CDAI< = 3.2) (%) | 480 (25.0) | 117 (17.7) | 363 (28.8) | < .001 |
| DAS28- REM (CDAI< = 2.6) (%) | 304 (15.8) | 77 (11.7) | 227 (18.0) | < .001 |
| ESR(mm/hr), Mean ± SD | 23.3 ± 20.6 | 25.3 ± 20.7 | 22.2 ± 20.5 | 0.013 |
| | N = 1779 | N = 621 | N = 1158 | |
| CRP (mg/L), Mean ± SD | 12.4 ± 20.6 | 14.5 ± 22.3 | 11.2 ± 19.4 | < .001 |
| | N = 1610 | N = 583 | N = 1027 | |
| HAQ-DI, Mean ± SD | 1.1 ± 0.70 | 1.1 ± 0.70 | 1.2 ± 0.70 | 0.528 |
| Presence of erosion (%) | 730 (46.7) | 128 (24.0) | 602 (58.4) | < .001 |
| | N = 1564 | N = 533 | N = 1031 | |
| Positive RF (%) | 1322 (73.3) | 436 (68.9) | 866 (74.0) | 0.002 |
| | N = 1804 | N = 633 | N = 1171 | |
| Number of main comorbidities, Mean ± SD | 1.1 ± 1.2 | 1.1 ± 1.1 | 1.1 ± 1.2 | 0.202 |
| Hypertension (%) | 688 (35.8) | 227 (34.4) | 461 (36.6) | 0.341 |
| CVD (%) | 217 (11.3) | 66 (10.0) | 151 (12.0) | 0.192 |
| Diabetes Mellitus (%) | 165 (8.6) | 61 (9.2) | 104 (8.3) | 0.463 |
| Lung diseases (%) | 262 (13.6) | 84 (12.7) | 178 (14.1) | 0.396 |
| Cancer (%) | 152 (7.9) | 52 (7.9) | 100 (7.9) | 0.965 |
| Depression (%) | 331 (17.2) | 123 (18.6) | 208 (16.5) | 0.241 |
| Gastrointestinal diseases | 335 (17.5) | 95 (14.4) | 240 (19.1) | 0.02 |
| Prior use of biologic (%) | 554 (28.9) | 63 (9.5) | 491 (39.0) | < .001 |
| Prior use of csDMARDs (%) | 1573 (81.9) | 355 (53.8) | 1218 (96.7) | < .001 |
| New bDMARDs start (%) | 304 (21.3) | 49 (8.8) | 255 (29.9) | < .001 |
| | N = 1409 | N = 555 | N = 854 | |
| New csDMARDs start (%) | 737 (52.2) | 424 (75.9) | 313 (36.7) | < .001 |
| | N = 1412 | N = 559 | N = 853 | |
| Current use of steroids (%) | 374 (19.5) | 155 (23.5) | 219 (17.4) | 0.001 |
| Current use of NSAIDs (%) | 477 (24.8) | 139 (21.1) | 338 (26.8) | 0.005 |

Numbers are presented as N and (%), unless indicated otherwise. RA, rheumatoid arthritis; RF, rheumatoid factor; ESR, erythrocyte sedimentation rate; CRP, C-reactive protein; HAQ-DI, Health Assessment Questionnaire-Disability Index; PtGA, patient global assessment; PhGA, physician global assessment; SJC28, swollen joint count-28; TJC28, tender joint counts-28; CDAI, clinical disease activity index; DAS28 ESR, Disease Activity Score 28-erythrocyte sedimentation rate; csDMARDs, conventional synthetic disease-modifying antirheumatic drugs; bDMARDs, biologic disease-modifying antirheumatic drugs; NSAID, non-steroidal anti-inflammatory drug; LDA: low disease activity; REM: remission

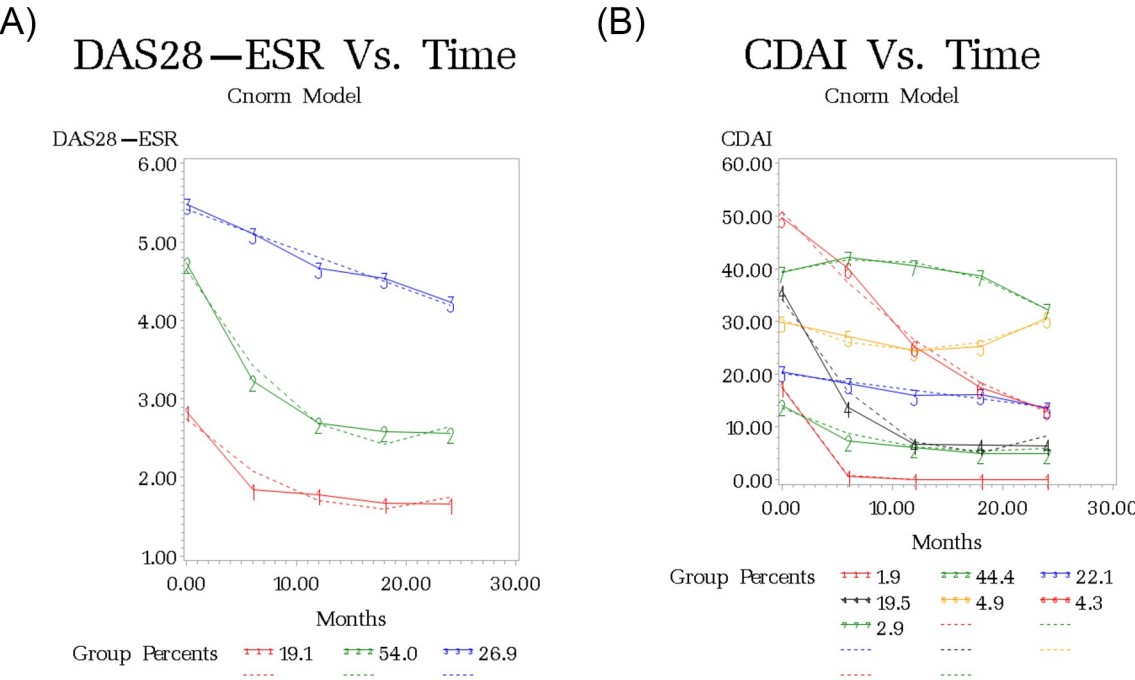

**Fig 2. Observed and fitted trajectories from latent growth curve analysis for disease course over 2 years in patients with early RA.**
A. DAS28-ESR. **DAS28-ESR category**: Remission: DAS-ESR< = 2.6; LDA: DAS-ESR< = 3.2; Moderate: 3.2<DAS28-ESR< = 5.0;
Severe: DAS28-ESR>5.0. B. CDAI. **CDAI category**: Remission (REM): CDAI< = 2.8; LDA: CDA< = 10; Moderate (MD): 10<CDAI<
= 22; Severe (HD): CDAI>22. Dashed line: fitted values. Solid line: observed values.

**CDAI.**    Using CDAI, seven subgroups were identified in patients with early RA (BIC:
-11713.4). Group 1 (MD-REM: 1.9%) with moderate disease at baseline rapidly achieved
remission at 6 months. Group 2 (MD-LDA: 44.4%) with moderate disease at baseline
improved gradually to LDA. Group 3 (MD-MD: 22.1%) with initial moderate disease activity
remained in a moderate state. Group 4 (HD-LDA: 19.5%) with initial high disease rapidly
improved to LDA. Group 5 (4.9%) and group 7 (2.9%) remained in high disease status over 2
years of follow-up (HD-HD: 7.8%). Group 6 with initial very high disease (VHD) rapidly
improved to low disease activity (LDA) state (VHD-LDA: 4.3%) (Fig 2B and Table A2 in S1
Appendix). Overall, using CDAI as a composite measure to describe activity in patients with

**Table 2. The mean (95% CI) DAS28-ESR values at each time point for trajectory classes in patients with early RA.**

| N = 660 | Group 1 | Group 2 | Group 2 |
|---|---|---|---|
| | **LDA-REM** | **MD-REM** | **HD-MD** |
| | **N = 110** | **N = 371** | **N = 179** |
| **Group percent** | **19.1%** | **54.0%** | **26.9%** |
| Baseline | 2.84 (2.29–3.19) | 4.73 (4.49–4.79) | 5.47 (5.25–5.57) |
| 6 months | 1.84 (1.83–2.33) | 3.22 (3.28–3.55) | 5.10 (4.98–5.23) |
| 12 months | 1.78 (1.48–1.94) | 2.69 (2.51–2.85) | 4.66 (4.69–4.91) |
| 18 months | 1.67 (1.37–1.82) | 2.59 (2.25–4.38) | 4.53 (4.38–4.62) |
| 24 months | 1.66 (1.47–2.48) | 2.56 (2.48–2.82) | 4.24 (4.04–4.34) |

**DAS28-ESR category**: Remission (REM): DAS-ESR< = 2.6; LDA: DAS-ESR< = 3.2; Moderate (MD):
3.2<DAS28-ESR< = 5.0; Severe (HD): DAS28-ESR>5.0

early RA showed that 30% of patients with moderate or high disease activity had no improvement over two years of follow-up (Group 3, 5, and 7). Of interest, using CDAI to identify disease course in early RA showed only 2% of patients reached remission (Fig 2B) within two years.

A cross tabulation for CDAI and DAS28 subgroups in patients with early RA is shown in Table A3 in S1 Appendix. Almost 60% of patients who were classified as MD-REM by CDAI were assigned to the LDA-REM group using DAS28, confirming disease remission at 24 months. Almost 90.0% of patients who were classified as VHD-MD by CDAI were assigned to the HD-MD group using DAS28, implying an improvement in diseases status using both measures.

### Disease trajectories in established RA

**DAS28-ESR.** Using DAS28-ESR, seven subgroups were identified in patients with established RA (BIC: -10000.81) (Fig 3A and Table A3 in S1 Appendix). Group 1 (REM-REM: 18.3%), group 2 (HD-REM: 14.2%), group 3 (LDA-LDA: 29.8%), group 4 (MD-MD: 18.1%), group 5 (HD-LDA: 10.3%), group 6 (HD-MD: 3.3%), and group 7 (HD-HD: 6.1%). Overall 27.8% of established RA patients with high disease activity showed an improvement in their disease status (group 2, 5, and 6) (Fig 3A and Table A3 in S1 Appendix).

**CDAI.** Using CDAI, seven subgroup of patients were also identified in patients with established RA (BIC: -22010.8) (Fig 3B and Table A5 in S1 Appendix). Group 1 (LDA-LDA: 37.1%), group 2 (MD-MD: 31.1%), group 3 (HD-HD: 8.9%), group 4 (HD-LDA: 10.9%), group 5 (HD-HD: 4.4%), group 6 (VHD -LDA: 7%), and group 7 (VHD-VHD: 0.6%). Only

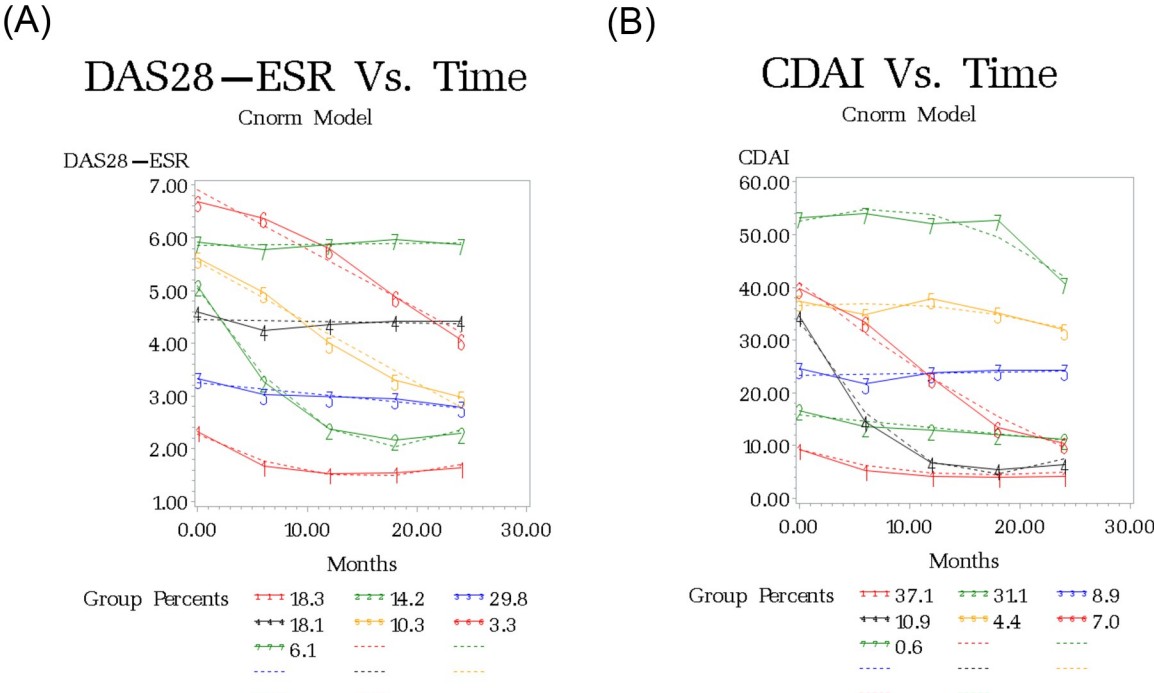

**Fig 3. Observed and fitted trajectories from latent growth curve analysis for disease course over 2 years in patients with established RA.** A. DAS28-ESR. **DAS28-ESR category**: Remission: DAS-ESR< = 2.6; LDA: DAS-ESR< = 3.2; Moderate: 3.2<DAS28-ESR< = 5.0; Severe: DAS28-ESR>5.0. B. CDAI. **CDAI category**: Remission: CDAI< = 2.8; LDA: CDA< = 10; Moderate: 10<CDAI< = 22; Severe: CDAI>22. Dashed line: fitted values. Solid line: observed values.

17.9% of patients with HD showed an improvement in their disease status (group 4 and 6) (Fig 3B and Table A5 in S1 Appendix).

A cross tabulation for CDAI and DAS28 subgroups in patients with established RA is shown in Table A6 in S1 Appendix. There were strong associations between subgroups identified by CDAI and DAS28 trajectories. Almost 97% of patients who remained in LDA at 24 months, based on CDAI (group 1), were assigned to the LDA or REM group (group 1, 2, 3, 5) using DAS28. Sixty-one percent of patients who were classified as HD-HD by CDAI (group 5) were also assigned to the HD-HD group using DAS28 (group 7). Seventy-five percent of patients in the VHD-VHD CDAI group (n = 8), were also assigned to the HD-HD DAS28 group.

### DAS28-ESR trajectories group characteristic in patients with early RA

Table 3 shows the sociodemographic, disease, and treatment profile of patients with early RA in three DAS28-ESR trajectories groups. Compared to the other two groups, patients in the HD-MD group (group 3) were significantly less likely to be married (66.5%, p = 0.03), and

**Table 3. Baseline characteristics of patients with early RA across DAS28-ESR trajectory groups.**

| N = 660 | Trajectories Group | | | P Value |
|---|---|---|---|---|
| | Group 1 | Group 2 | Group 3 | |
| | LDA-REM (N = 110) | MD-REM (N = 371) | HD-MD (N = 179) | |
| Female (%) | 74 (67.3) | 272 (73.3) | 141 (78.8) | 0.093 |
| Age, years, Mean ± SD | 54.7 ± 12.5 | 56.8 ± 13.5 | 57.5 ± 13.3 | 0.210 |
| Marital status, married (%) | 89 (80.9) | 263 (70.9) | 119 (66.5) | **0.030** |
| Post-secondary education (%) | 69 (62.7) | 223 (60.1) | 87 (48.6) | **0.017** |
| Current smoker (%) | 17 (15.5) | 56 (15.1) | 37 (20.7) | 0.277 |
| HAQ-DI, Mean ± SD | 0.6 ± 0.5 | 1.1 ± 0.7 | 1.6 ± 0.6 | **< .001** |
| HAQ-pain, Mean ± SD | 0.9 ± 0.7 | 1.4 ± 0.8 | 1.9 ± 0.8 | **< .001** |
| ESR (mm/hr), Mean ± SD | 9.8 ± 9.2 | 26.4 ± 19.4 | 32.6 ± 23.3 | **< .001** |
| CRP (mg/L) Mean ± SD | 5.3 ± 12.2 | 15.7 ± 23.1 | 17.8 ± 23.9 | **< .001** |
| Positive RF (%) | 72 (65.5) | 253 (68.2) | 111 (62.0) | 0.648 |
| Number of main comorbidities, Mean ± SD | 0.8 ± 1.1 | 1.0 ± 1.1 | 1.3 ± 1.2 | **< .001** |
| Hypertension (%) | 34 (30.9) | 119 (32.1) | 74 (41.3) | 0.071 |
| CVD (%) | 11 (10.0) | 38 (10.2) | 17 (9.5) | 0.963 |
| Diabetes Mellitus (%) | 5 (4.5) | 35 (9.4) | 21 (11.7) | 0.121 |
| Lung diseases (%) | 12 (10.9) | 42 (11.3) | 30 (16.8) | 0.165 |
| Cancer (%) | 9 (8.2) | 27 (7.3) | 16 (8.9) | 0.788 |
| Depression (%) | 10 (9.1) | 63 (17.0) | 50 (27.9) | **< .001** |
| Gastrointestinal diseases | 12 (10.9) | 54 (14.6) | 29 (16.2) | 0.46 |
| Prior use of bDMARDs (%) | 11 (10.0) | 27 (7.3) | 25 (14.0) | **0.043** |
| Prior use of csDMARDs (%) | 75 (68.2) | 176 (47.4) | 104 (58.1) | **< .001** |
| New bDMARDs start (%) | 5 (4.5) | 22 (5.9) | 22 (12.3) | **0.021** |
| New csDMARDs start (%) | 52 (47.3) | 256 (69.0) | 116 (64.8) | **0.003** |
| Current use of steroids (%) | 155 (23.5) | 9 (8.2) | 88 (23.7) | **< .001** |

**DAS28-ESR category**: Remission (REM): DAS-ESR< = 2.6; LDA: DAS-ESR< = 3.2; Moderate (MD): 3.2<DAS28-ESR< = 5.0; Severe (HD): DAS28-ESR>5.0

Numbers are presented as N and (%), unless indicated otherwise. CDAI, clinical disease activity index; PtGA, patient global assessment; PhGA, physician global assessment; SJC28, swollen joint count-28; TJC28, tender joint counts-28; DAS28 ESR, Disease Activity Score 28-erythrocyte sedimentation rate; HAQ-DI, Health Assessment Questionnaire-Disability Index; RF, rheumatoid factor; bDMARDs, biologic disease-modifying antirheumatic drugs csDMARDs, conventional synthetic disease-modifying antirheumatic drugs.

have post secondary education (48.6%, p = 0.02) at enrolment. Physical function measured by HAQ-DI (mean = 1.6, p<0.001) and patient reported pain (mean = 1.9, p<0.001) was also significantly worse in these patients compared to the other two groups. The mean number of main comorbidities was significantly higher in this trajectory group (mean = 1.3, p<0.001). A significantly higher proportion of patients in this group used prior bDMARDs (14%, p = 0.04), started new bDMARDs at enrolment (12.3%, p = 0.02), and were currently using steroids (23.7%, p<0.001) compared to groups 1 and 2.

Compared to the other two groups, a lower proportion of patients in group 1 (LDA-REM) started a new traditional DMARD at enrolment (47.3%, p = 0.003) and were more likely to be using them before enrolment (68.2%, p<0.001). No significant differences in age, gender, rheumatoid factor positivity, and current smoking status were found between the three groups (Table 3).

## Discussion

In this study we used two composite measures of disease activity to look at disease course in early and established RA patients enrolled in the OBRI. Using DAS28-ESR and CDAI, we detected seven discrete trajectories for established RA categorizing patients' disease activity at the time of registry enrolment and two years later.

Using DAS28-ESR we found that almost 27.8% of patients experienced some degree of improvement, while one-fourth (24.2%) did not show any improvement. Using CDAI to determine disease course in established patients resulted in similar patterns, however only 17.9% showed improvement. The lack of response in this group of established RA patients may be the result of inappropriate treatment strategies as well as varied comorbidity profiles compared to early RA.

Only 40% of early RA and 30% of established RA patients reached either REM or LDA within 6 months. While the assessment of treat-to-target strategies is beyond the scope of this study, these findings suggest that further investigations are required to better understand why most patients are not reaching the primary target (i.e., LDA or REM) after 6 months of treatment. Perhaps certain clusters or subgroups of RA patients, for example those with more comorbidities, require more aggressive treatment strategies or it is possible that rheumatologist are using other health outcomes (i.e., not DAS and CDAI scores) to assess improvements.

One study from the British Society for Rheumatology Biologics Register for RA (BSRBR-RA) identified four district trajectories (maximal response HD-REM: 8.7%; substantial response HD-LDA: 32%; modest response HD-MD: 55%; minimal response HD-HD: 4.5%) for patients with RA after TNFi initiation [3]. Disease activity was defined as severe (mean DAS28-ESR: 6.5) for the whole cohort as a requirement for starting a biologic treatment in the UK. As they did not present the results by disease duration, a direct comparison with our results was not possible. However, if we compare the HD-REM cohorts, our established cohort showed a greater response (Group 2 in established RA: HD-REM; 14.2%) compared to their maximal response group (HD-REM; 8.7%) [3]. The minimal response rate (HD-HD: 4.5%) derived form their analysis [3] was almost similar to group 7 (HD-HD: 6.1%) in our established RA cohort. Another recent study from UK, using 4 component DAS28-CRP also found three trajectories among 2991 patients with baseline means DAS 5.6 and disease duration of 10 years during 12 months follow-up (rapid responders: 67%, gradual responders: 30.7% and poor responders: 2.3%) [8].

A study from the DREAM-RA registry [9] investigated disease course over two years of follow-up in 180 patients with established RA (categorised into two groups based on their pain phenotype). Compared to our study (DAS28-ESR mean in established RA: 4.1), disease activity

measured by DAS28 was lower in their group (the mean values for the DAS28 in the non-nociceptive and nociceptive pain groups at baseline were 2.8 and 2.1, respectively). In terms of disease course during follow-up, they showed no significant change in DAS28 scores over time for the total cohort. One possible explanation for the lack of subgroup trajectories in their study is that all patients included in their cohort were in LDA/remission state at baseline and remained the same during two years, suggesting that the established patients (mean disease duration 8 year) [9] in the DREAM-RA registry were all well managed.

Using data from nine different national registries, another study identified different groups of trajectories following a new biologic treatment [2]. The mean disease duration for 3898 patients included in the study was 12 years. They identified three discrete groups of patients: 1) gradual responders (91.7%) with a baseline mean DAS28 of 4.1; 2) rapid responders (5.6%) with baseline DAS28 of 5.8; and 3) inadequate responders (2.6%) with at baseline DAS28 of 5.1. Compared to our stablished cohort which showed almost 30% improvement from HD to LDA or remission for DAS28-ESR, none of the identified groups in this study reached LDA or remission over two years of follow-up and only 6% of patients with high disease at baseline reached moderate disease status [2]. Heterogeneity of treatment strategy and reimbursement policy across the nine different countries in this pooled analysis may explain the inconsistent findings.

In our study we found that compared to established RA, the disease courses and number of trajectory subgroups identified was different for patients with early RA, where more than 70% showed improvement over two years of follow-up using CDAI and 100% using DAS28-ESR. Barnabe et al. 2015 [1] identified 5 subgroup of 1586 patients with early RA across Canada, with all showing some degree of improvement over two years of follow-up. However, the disease severity in our study cohort was slightly lower (mean of DAS28-ESR: 4.6 vs. 5.1) [1]. Almost half of their patients were in high disease status at cohort inception, while only 12% of early RA patients in our study showed high disease activity suggesting most patients enrolled in the OBRI have been well managed. Another explanation for this difference might be the definition used for early RA in these two studies. In our registry, we define early RA as disease duration less than 2 years whereas they defined early RA as less than 1 year.

Similar to our study, Siemons et al. 2014 [10] identified three trajectories in patients with early RA following a treat to target strategy over 1 year of follow-up. They found 83% with fast decreasing disease activity and stabilizing remission at 9 months. These results are comparable to our findings where 19% and 54% of early RA patients were in stable DAS28-ESR remission at 6 and 12 months, respectively (Table 2). RA_Map Consortium also identified three DAS28-CRP trajectory classes in 267 untreated RA patients from 18 UK centres; 21.7% as inadequate responders, 21.3% as higher baseline activity and 57% as lower baseline activity (moderate status at baseline) both with sustained improvement over 18 months. Lower HAQ-DI, better mental wellbeing, use of dual RA medication at baseline, alcohol consumption, and being female was associated with lower DAS-CRP over time [11]. Their mean DAS28-CRP at baseline was similar to our cohort [4.85 (SD: 0.84)] indicating similar disease course over time (3 trajectory groups in early RA) between the two studies. The RA-Map consortium in another study, conducted in the UK found three DAS28 trajectory classes among 3290 patients from non-biologic arms of phase II and III clinical trials between 2002 and 2012. Latent class mixed model identified differential non–biologic response with three trajectory subpopulations in both MTX-naïve and MTX-exposed patients [12].

In our study we also found that using DAS28-ESR identified fewer subgroups of early RA patients (three discrete trajectories) compared to using CDAI (seven discrete trajectories). Furthermore, by using CDAI, 30% of patients with early RA did not show any degree of improvement, including remission, whereas using DAS28-ESR all patients showed improvement. The

presence of patient global assessment (PtGA) as one of the components for CDAI may explain this difference. In another recent study [13], we found that agreement in the classification of LDA/remission between CDAI (≤10) and DAS28-ESR (≤3.2) was fair to moderate, while agreement in the classification of remission between CDAI (≤2.8) and DAS28 (≤2.6) was poor to fair. PtGA also showed the lowest correlation with the remaining CDAI components which became gradually lower towards lower CDAI disease scores [13]. Other studies have shown low agreement between PtGA, joint counts and markers of inflammation especially when ACR/EULAR Boolean remission was not obtained, and that PtGA remained high compared to joint counts and other markers of disease activity [14, 15]. Nevertheless a cross-tabulation between subgroups identified by CDAI and DAS28 showed some association between these subgroups. A stronger association between subgroups of CDAI and DAS28 was also shown in established RA.

Using CDAI for remission makes it more difficult to show improvements in health outcomes, however, its use is more practical in a clinical setting as it does not require laboratory measures (ESR or CRP).

We additionally compared baseline characteristics between the three subgroups of trajectories which were identified in the early RA cohort using DAS28-ESR. We showed that being married, having post secondary education, lower HAQ-DI, lower patient reported pain, lower ESR, and fewer comorbidities were predictive of reaching remission. Siemon et al. 2014 [10] also compared these characteristics between trajectories and found only male sex was a predictive factor for a fast response. Barnabe et al. 2015 [1] found that patients showing the largest improvement (HD-REM) and the best prognosis (MD-REM) are less than 50 years old and have less comorbidity. Similar to our study they also showed that patients starting with high disease are more likely to have lower levels of education [1]. Norton et al. 2014 [16] described sociodemographic differences between trajectory groups and found that groups with the highest level of HAQ-DI were more likely to have higher comorbidity scores, and lower education, social class, and employment level.

In an exploratory analysis we also found meaningful associations between DAS28 subgroups and improvement in functional disability (HAQ-DI and HAQ-pain) at 12 and 24 months follow-up (Table A7 in S1 Appendix) which contributes the validity of subgroup trajectories developed in this study.

We used LCGM, as the most common developed approach, to show patient's disease course and heterogeneity between subjects over time. This approach has been previously used by other studies in the field of rheumatic disease to identify disease course [1, 3, 10, 17] and swollen joint count trajectories in juvenile inflammatory arthritis [18]. Using clinical data from two Canadian pediatric rheumatology centers they identified five trajectory groups with significant differences in the international League of Associations for Rheumatology categorizations (ILAR) [18].

One of the limitations of longitudinal studies is lost to follow-up. The cohort included in our analysis had complete data for two years of follow-up after baseline, i.e., at 6, 12, 18, and 24 months, which can be considered a strength. However, unmeasured variables in our study could explain some of the heterogeneity seen between trajectory groups. Another limitation for this study is the possibility of selection bias as we applied several inclusion and exclusion criteria for our patients, therefore the results may not generalizable to other RA population. There is also a possibility of more clinic visits by patients with high disease activity compared to those with LDA status, which may affect the impression of disease course toward persistent high disease activity in these patients.

In conclusion disease course is different in early and established RA. After 2 years of follow-up, only 14.2% of established RA patients reached DAS28-ESR remission compared to

73.1% of early RA patients. When CDAI was used as a measure of disease activity, none of the established RA patients reached remission and only 2% of the early RA patients reached remission over 2 years' follow-up. This may reflect the impact of the PtGA component on CDAI as a composite measure for disease activity. The findings also suggest that sociodemographic characteristics and early treatment impact disease course.

## Supporting information

**S1 Appendix.**
(DOCX)

## Acknowledgments

We would like to thank all OBRI participants and following investigators: Drs. Ahluwalia, V., Ahmad, Z., Akhavan, P., Albert, L., Alderdice, C., Aubrey, M., Aydin, S., Bajaj, S., Bell, M., Bensen, W., Bhavsar, S., Bobba, R., Bombardier, C., Bookman, A., Brophy, J., Cabral, A., Carette, S., Carmona, R., Chow, A., Choy, G., Ciaschini, P., Cividino, A., Cohen, D., Dixit, S., Faraawi, R., Haaland, D., Hanna, B., Haroon, N., Hochman, J., Jaroszynska, A., Johnson, S., Joshi, R., Kagal, A., Karasik, A., Karsh, J., Keystone, E., Khalidi, N., Kuriya, B., Lake, S., Larche, M., Lau, A., LeRiche, N., Leung, Fe., Leung, Fr., Mahendira, D., Matsos, M., McDonald-Blumer, H., McKeown, E., Midzic, I., Milman, N., Mittoo, S., Mody, A., Montgomery, A., Mulgund, M., Ng, E., Papneja, T., Pavlova, V., Perlin, L., Pope, J, Purvis, J., Rai, R., Rohekar, G., Rohekar, S., Ruban, T., Samadi, N., Sandhu, S., Shaikh, S., Shickh, A., Shupak, R., Smith, D., Soucy, E., Stein, J., Thompson, A., Thorne, C., Wilkinson, S.

Dr Claire Bombardier leads OBRI investigators group with following email contact: claire.bombardier@utoronto.ca

| Investigator | Affiliation |
|---|---|
| Dr. Carter Thorne | Assistant Professor, University of Toronto, Toronto, ON, Canada |
| Dr. Janet Pope | Professor of Medicine, Western University, Schulich School of Medicine & Dentistry, London, ON, Canada |
| Dr. Alfred Cividino | Professor Emeritus, McMaster University, Hamilton, ON, Canada |
| Dr. Jane Purvis | Courtesy staff Peterborough Regional Health Centre, Peterborough, ON, Canada Queens University Adjunct Faculty, Department of Family Medicine, Kingston, ON, Canada |
| Dr. Vandana Ahluwalia | Division of Rheumatology, Department of Internal Medicine, William Osler Health System, Brampton, ON, Canada |
| Dr. Sangeeta Bajaj | William Osler Health System, Brampton, ON, Canada University of Toronto, Toronto, ON, Canada McMaster University, Hamilton, ON, Canada |
| Dr. Arthur Karasik | Community practice, Toronto, ON, Canada |
| Dr. Andrew Chow | Assistant Clinical Professor Medicine, McMaster University, Hamilton, ON, Canada |
| Dr. Brian Hanna | Community practice, Cambridge, ON, Canada |
| Dr. Catherine Alderdice | Community practice, Guelph, ON, Canada |
| Dr. Nader Khalidi | St. Joseph's Healthcare/McMaster University, Hamilton, ON, Canada |
| Dr. Ali Shickh | Community practice, Bowmanville, ON, Canada |
| Dr. Frances Leung | Community practice, Toronto, ON, Canada |
| Dr. Bindee Kuriya | Sinai Health System, University of Toronto, Toronto, ON, Canada |
| Dr. Edward Keystone | Professor Emeritus, University of Toronto, Toronto, ON, Canada |

*(Continued)*

(Continued)

| Investigator | Affiliation |
|---|---|
| Dr. Jacqueline Hochman | Women's College Hospital, Toronto, ON, Canada |
| Dr. Claire Bombardier | Toronto General Research Institute, University Health Network, Toronto, ON<br>Mount Sinai Hospital, Toronto, ON<br>Division of Rheumatology, Faculty of Medicine, University of Toronto, Toronto, ON<br>IHPME, Dalla Lana School of Public Health, University of Toronto, Toronto, ON, Canada |
| Dr. Pooneh Akhavan | Mount Sinai Hospital, Toronto, ON, Canada |
| Dr. Elaine Soucy | Trillium Health Partners, Mississauga, ON, Canada<br>Lecturer (adjunct) University of Toronto, Toronto, ON, Canada |
| Dr. Felix Leung | Community practice, Toronto, ON, Canada |
| Dr. Ami Mody | Community practice, Mississauga, ON, Canada |
| Dr. Angela Montgomery | Community practice, Ottawa, ON, Canada |
| Dr. Michael Aubrey | Markham Stouffville Hospital, Markham, ON, Canada |
| Dr. E. Ng Tung Hing | Centre of Arthritis Excellence, Newmarket, ON, Canada<br>TAP research program, Newmarket, ON, Canada |
| Dr. Heather McDonald-Blumer | Division of Rheumatology, Mount Sinai Hospital, Toronto, ON, Canada<br>University Health Network, Toronto, ON, Canada |
| Dr. Zareen Ahmad | Toronto Scleroderma Program, Division of Rheumatology, Toronto Western and Mount Sinai Hospitals, Toronto, ON, Canada<br>Department of Medicine, University of Toronto, Faculty of Medicine, Toronto, ON, Canada |
| Dr. Mark Matsos | Associate Professor, McMaster University, Hamilton, ON, Canada |
| Dr. Raj Carmona | Associate Clinical Professor, Director, Medical Foundation 4, MD Program, McMaster University, Hamilton, ON, Canada |
| Dr. Shikha Mittoo | Community practice, Toronto, ON, Canada |
| Dr. Allan Kagal | Mackenzie Health, Richmond Hill, ON, Canada |
| Dr. Sankalp Bhavsar | McMaster University, Hamilton, ON, Canada |
| Dr. Arthur Bookman | Professor Medicine, University of Toronto, Staff Rheumatologist, University Health Network, Toronto, ON, Canada |
| Dr. Lori Albert | Faculty of Medicine, University of Toronto, Toronto, ON, Canada<br>Rheumatology Clinic, Toronto Western Hospital, University Health Network, Toronto, ON, Canada |
| Dr. Saeed Shaikh | Assistant Clinical Professor, McMaster University, Hamilton, ON, Canada |
| Dr. Jonathan Stein | Department of Medicine, University of Toronto, Toronto, ON, Canada<br>Department of Medicine, St Joseph's Health Centre, Toronto, ON, Canada |
| Dr. Nicole le Riche | Department of Medicine, Western University, London, ON, Canada<br>Division of Rheumatology, St. Joseph's Health Care, London, ON, Canada |
| Dr. Andy Thompson | Associate Professor of Medicine, Western University, London, ON, Canada |
| Dr. Gina Rohekar | University of Western Ontario, London, ON, Canada |
| Dr. Sherry Rohekar | Associate Professor, Division of Rheumatology, Department of Medicine, Western University, London, ON, Canada |
| Dr. William Bensen | St. Joseph's Healthcare, McMaster University, Hamilton, ON, Canada |
| Dr. Viktoria Pavlova | Community practice, Hamilton, ON, Canada |
| Dr. Sanjay Dixit | Assistant Clinical Professor, Department of Medicine, Division of Rheumatology, McMaster University, Hamilton, ON, Canada |
| Dr. Manisha Mulgund | Community practice, Hamilton, ON, Canada |
| Dr. Dana Cohen | Community practice, Vaughan, ON, Canada |
| Dr. Patricia Ciaschini | Community practice, Sault Ste. Marie, ON, Canada |
| Dr. Simon Carette | Division of Rheumatology, Mount Sinai Hospital, University of Toronto, Toronto, ON, Canada |

(*Continued*)

(Continued)

| Investigator | Affiliation |
|---|---|
| Dr. Sindhu Johnson | Division of Rheumatology, Toronto Western Hospital and Mount Sinai Hospital, Toronto, ON, Canada<br>Institute of Health Policy, Management and Evaluation, University of Toronto, Toronto, ON, Canada |
| Dr. Nigil Haroon | Schroeder Arthritis Institute, Krembil Research Institute, University Health Network, and University of Toronto, Toronto, ON, Canada |
| Dr. Nooshin Samadi | Community practice, Newmarket, ON, Canada |
| Dr. Louise Perlin | St. Michael's Hospital, Toronto, ON, Canada |
| Dr. Rachel Shupak | University of Toronto, Toronto, ON, Canada |
| Dr. Dharini Mahendira | Division of Rheumatology, Department of Medicine, University of Toronto, Toronto, ON, Canada |
| Dr. Thanu Ruban | Department of Medicine, Markham Stouffville Hospital, Markham, ON, Canada |
| Dr. Raja Bobba | Faculty of Health Sciences, McMaster University, Hamilton, ON, Canada |
| Dr. Rajwinder Dhillon | Assistant Clinical Professor (Adjunct), McMaster University, Hamilton, ON, Canada |
| Dr. Douglas Smith | Associate Professor, Division of Rheumatology, Department of Medicine, University of Ottawa, Ottawa, ON, Canada |
| Dr. Jacob Karsh | Rheumatology, University of Ottawa Faculty of Medicine, Ottawa, ON, Canada |
| Dr. Anna Jaroszynska | Community practice, Burlington, ON, Canada |
| Dr. Derek Haaland | Rheumatologist, Clinical Immunologist & Allergist, Medical Director, The Waterside Clinic, Barrie, ON, Canada<br>Associate Clinical Professor, McMaster University, Hamilton, ON, Canada<br>Assistant Professor, Northern Ontario School of Medicine, Laurentian University Campus, Sudbury, ON, Canada |
| Dr. Arthur Lau | Department of Medicine, Division of Rheumatology, McMaster University, Hamilton, ON, Canada |
| Dr. Maggie Larche | Professor, Department of Medicine, McMaster University, Hamilton, ON, Canada |
| Dr. Raman Joshi | Brampton Civic Hospital, William Osler Health Systems, Brampton, ON, Canada |
| Dr. Tripti Papneja | William Osler Health System, Brampton, ON, Canada |
| Dr. Antonio Cabral | Division of Rheumatology, Department of Medicine, The Ottawa Hospital, University of Ottawa, Ottawa, ON, Canada |
| Dr. Sibel Aydin | Professor, Faculty of Medicine, Ottawa Hospital Research Institute, University of Ottawa, Ottawa, ON, Canada |
| Dr. Ines Midzic | Division of Rheumatology, Department of Medicine, University of Ottawa, Ottawa, ON, Canada |
| Dr. Nataliya Milman | University of Ottawa, The Ottawa Hospital, The Ottawa Hospital Research Institute, Ottawa, ON, Canada |
| Dr. Rafat Faraawi | McMaster University, Hamilton, ON, Canada |
| Dr. Julie Brophy | Community practice, Guelph, ON, Canada |
| Dr. Mary Bell | Sunnybrook Health Sciences Centre, Department of Medicine, Division of Rheumatology, Associate Scientist, Sunnybrook Research Institute, Toronto, ON, Canada<br>Professor of Medicine, University of Toronto, Toronto, ON, Canada |
| Dr. Gregory Choy | Division of Rheumatology, Sunnybrook Health Sciences Centre, Toronto, ON, Canada<br>Assistant Professor, University of Toronto, Toronto, ON, Canada |
| Dr. Sharron Sandhu | Division of Rheumatology, Sunnybrook Health Sciences, Toronto, ON, Canada<br>University of Toronto, Toronto, ON, Canada |
| Dr. Emily McKeown | Sunnybrook Health Sciences Centre, Division of Rheumatology, Toronto, ON, Canada |
| Dr. Shirley Lake | Sunnybrook Health Sciences Centre, Sunnybrook Research Institute, Toronto, ON, Canada |
| Dr. Tooba Ali | Assistant Professor, Department of Medicine, Queen's University, Kingston, ON, Canada |
| Dr. Saara Rawn | Algoma District Medical Group, Sault Ste. Marie, ON, Canada<br>Assistant Professor, Northern Ontario School of Medicine, Thunder Bay, ON, Canada |
| Dr. Raman Rai | Community practice, Brampton, ON, Canada |

## Author Contributions

**Conceptualization:** Claire Bombardier.

**Data curation:** Mohammad Movahedi, Xiuying Li.

**Formal analysis:** Mohammad Movahedi.

**Investigation:** Mohammad Movahedi, Claire Bombardier.

**Methodology:** Mohammad Movahedi.

**Project administration:** Angela Cesta.

**Resources:** Xiuying Li, Claire Bombardier.

**Software:** Mohammad Movahedi.

**Writing – original draft:** Mohammad Movahedi, Angela Cesta.

**Writing – review & editing:** Mohammad Movahedi, Angela Cesta, Xiuying Li, Claire Bombardier.

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
