## [Decision Letter · Decision Letter 0]

6 Apr 2022

PONE-D-22-02930Disease Activity Trajectories for Early and Established Rheumatoid Arthritis:  Real-World Data from a Rheumatoid Arthritis CohortPLOS ONE

Dear Dr. Movahedi,

Thank you for submitting your manuscript to PLOS ONE. After careful consideration, we feel that it has merit but does not fully meet PLOS ONE’s publication criteria as it currently stands. Therefore, we invite you to submit a revised version of the manuscript that addresses the points raised during the review process.

All reviewers found some interests in this study, but also pointed out a number of comments and criticisms, which require amendment or improvement before publication. I ask the authors to fully respond to all comments made by reviewers in the revised manuscript. 

We look forward to receiving your revised manuscript.

Kind regards,

Masataka Kuwana, MD, PhD

Academic Editor

PLOS ONE

Journal Requirements:

“OBRI was funded by peer-reviewed grants from CIHR (Canadian Institute for Health Research), Ontario Ministry of Health and Long-Term Care (MOHLTC), Canadian Arthritis Network (CAN) and unrestricted grants from: Abbvie, Amgen, Aurora, Bristol-Meyers Squibb, Celgene, Gilead, Hospira, Janssen, Lilly, Medexus, Merck, Novartis, Pfizer, Roche, Sandoz, Sanofi, & UCB Acknowledgment: Dr. Bombardier held a Canada Research Chair in Knowledge Transfer for Musculoskeletal Care and a Pfizer Research Chair in Rheumatology”

“The authors received no specific funding for this work.”

5. One of the noted authors is a group or consortium “OBRI investigators”. In addition to naming the author group, please list the individual authors and affiliations within this group in the acknowledgments section of your manuscript. Please also indicate clearly a lead author for this group along with a contact email address.

Reviewers' comments:

Reviewer's Responses to Questions

**Comments to the Author**

1. Is the manuscript technically sound, and do the data support the conclusions?

Reviewer #1: Partly

Reviewer #2: Partly

Reviewer #3: No

2. Has the statistical analysis been performed appropriately and rigorously? 

Reviewer #1: Yes

Reviewer #2: Yes

Reviewer #3: No

3. Have the authors made all data underlying the findings in their manuscript fully available?

Reviewer #1: Yes

Reviewer #2: No

Reviewer #3: No

4. Is the manuscript presented in an intelligible fashion and written in standard English?

Reviewer #1: Yes

Reviewer #2: Yes

Reviewer #3: Yes

5. Review Comments to the Author

Reviewer #1: The authors analyzed the disease course of early and established RA according to trajectories group identified by LCGM. They found that different measure of disease activity may show different pattern of disease course; specifically lower remission rates in CDAI than in DAS28-ESR over 2-years follow up. The reviewer think that the authors need to add the additional analysis and discussion as listed below.

1. The reviewer could not find the definition of major comorbidities in this study. Please add the definition of major comorbidities in the method section.

2. The authors just compared the number of main comorbidities between early RA and established RA groups in table 1. The authors should add the comparisons of each comorbidity based on the detail of the above definition between early RA and established RA groups.

3. Because the authors found the significant difference about remission rates based on DAS28-ESR or CDAI in an identical early RA cohort, the authors should discuss which is better to use DAS28-ESR or to use CDAI in clinical practice to prevent radiographic progression or to achieve better outcome of patients’ quality of life.

4. In table 3, please add the comparisons of each comorbidity based on the detail of the above definition.

5. In the results section, please add the analysis for CDAI trajectories group characteristics in patients with early RA.

6. In the results section, please add the analysis for DAS28-ESR trajectories group characteristics in patients with established RA.

7. In the results section, please add the analysis for CDAI trajectories group characteristics in patients with established RA.

8. Please add the discussion about selection bias as one of the limitation of the study because a significant proportion of patients were excluded at the final analysis as shown in Figure 1.

Reviewer #2: I have a number of comments regarding this paper

1. Literature review: There are a number of papers that have performed similar analyses to the one in this paper which have not been cited. For example, Dagliati et al, A&R (2020); RA-MAP Consortium, Ther Adv in MusculSkelet Dis (2021); RA-MAP Consortium, RMD Open, 2018; McWilliams et al, AR&T (2016). These should be added. It would also be useful for the authors to explain what they work will add to that which is out there already.

2. Definition of LDA, MD, HD, Remission: The authors should provide in the main text of the paper the definitions for LDA, MD, HD, Remission with respect to DAS28-ESR or CDAI, rather than relegating only to the footnotes under Figures 2b, 3a

and 3b.

3. Validation: There is an opportunity here to perform some form of validation of the clustering/subgroup structures found using DAS28-ESR and CDAI across early and established RA either for example through some form of cross-validation or by splitting the data into a testing and training set. My concern is that the clusters/subgroups found are not necessarily robust either across similar populations or even within the same population, especially the very small clusters/subgroups. These subgroups would be more meaningful/convincing if the authors could show at the least that they are stable or they associate meaningfully to later health outcomes such as damage progression (through use of X-rays) or functional disability.

4. Given that the number and presumably the clusters themselves may be different depending on whether DAS28-ESR or CDAI is used, it would be helpful to present the cross-tabulations of the clustering structure based on DAS28-ESR and the clustering structure based on CDAI for both early and established RA, so as to understand how these structure differ. Also it would be helpful for the authors to discuss what the implications of these differing clustering structures on treat-to-target strategies which aim to improve health outcomes of patients with RA.

5. Statistical Methods: The authors should provide more details on how they assign patients to subgroups based on the latent growth curve model (LCGM or LGCM?).

6. In Figures 2a, 2b, 3a and 3b , the authors should make the figures clearer. For example, (i) explicitly state that the solid lines correspond to the observed and the dashed lines to the expected; (ii) add a legend indicating that Group 1 corresponds to the red lines, Group 2 to the green lines and Group 3 to the blue lines, and make it clear that the numbers alongside the lines are the percentage of patients within these groups; and (iv) add the confidence bands around the curves to reflect the uncertainty.

7. The actual results (estimates and standard errors) from the final latent growth curve models should be provided in the Supplementary.

8. The authors need to discuss the possible impact selection bias has on their findings given that they only look at patients with complete outcome data (DAS28-ESR and CDAI) at baseline and all follow-up visits up to 24-months.

Reviewer #3: Dear authors

Thank you for an interesting paper that attempts to identify the trajectories of disease activity in a large cohort of patients with rheumatoid arthritis. I think that the paper reads well.

I have some concerns regarding the methods and presentation of data.

Major concerns:

1. I find it odd that all patients with early RA can be categorised into just three categories of DAS28. Were there no patients who remained in high disease activity for the duration of the two years? Similarly, did no patients go from MD to HD in the established RA group? To this end it would be useful to include either standard deviations or IQR in table 2.

2. I missed tables similar to table 2 and 3 for the other categorisations? If there is not enough space in the manuscript, then included them in the appendix.

3. Were there some patients who participated in borth early and late

4. Although telephone calls are made every 6-months, CDAI and DAS28 cannot be collected by telephone. How often were the routine care visits and is it possible that routine care visits were initiated by patients who experienced high disease activity, thus giving a false impression of persistently high disease activity?

Minor

1.The abstract states that "Only 14.2% of established RA reached DAS28-ESR remission" whereas the discussion states "Using DAS28-ESR we found that 48% of established patients were in remission and low disease status at

enrolment and retained their disease status over two years’ follow-up. Almost 27.8% of patients

experienced some degree of improvement". While both may be true I think there should be a better alignment between the message given in the abstract and in the discussion.

2. What was the average number of visits in each group? It would be helpful if you could show the number of patients with visit at each time point in each trajectory to help the reader judge the validity of the information.

6. PLOS authors have the option to publish the peer review history of their article (what does this mean?). If published, this will include your full peer review and any attached files.

Reviewer #1: No

Reviewer #2: No

Reviewer #3: No

---

## [Author Response · Author response to Decision Letter 0]

2 Jun 2022

We have provided answers to reviewers and uploaded related file.

---

## [Decision Letter · Decision Letter 1]

19 Jul 2022

PONE-D-22-02930R1Disease Activity Trajectories for Early and Established Rheumatoid Arthritis:  Real-World Data from a Rheumatoid Arthritis CohortPLOS ONE

Dear Dr. Movahedi,

Thank you for submitting your manuscript to PLOS ONE. After careful consideration, we feel that it has merit but does not fully meet PLOS ONE’s publication criteria as it currently stands. Therefore, we invite you to submit a revised version of the manuscript that addresses the points raised during the review process.

Our reviewers think that some of the critical comments have not been adequately answered in the revised version. I ask the authors to respond to the points raised by reviewers in the re-revised version.

We look forward to receiving your revised manuscript.

Kind regards,

Masataka Kuwana, MD, PhD

Academic Editor

PLOS ONE

Reviewers' comments:

Reviewer's Responses to Questions

**Comments to the Author**

1. If the authors have adequately addressed your comments raised in a previous round of review and you feel that this manuscript is now acceptable for publication, you may indicate that here to bypass the “Comments to the Author” section, enter your conflict of interest statement in the “Confidential to Editor” section, and submit your "Accept" recommendation.

Reviewer #1: (No Response)

Reviewer #2: (No Response)

Reviewer #3: All comments have been addressed

2. Is the manuscript technically sound, and do the data support the conclusions?

Reviewer #1: No

Reviewer #2: Partly

Reviewer #3: Yes

3. Has the statistical analysis been performed appropriately and rigorously? 

Reviewer #1: No

Reviewer #2: Yes

Reviewer #3: Yes

4. Have the authors made all data underlying the findings in their manuscript fully available?

Reviewer #1: No

Reviewer #2: No

Reviewer #3: No

5. Is the manuscript presented in an intelligible fashion and written in standard English?

Reviewer #1: Yes

Reviewer #2: Yes

Reviewer #3: Yes

6. Review Comments to the Author

Reviewer #1: (No Response)

Reviewer #2: 1. The authors have not addressed (or attempted to address) my concern regarding validation (Comment 3). As mentioned in my earlier review, "My concern is that the clusters/subgroups found are not necessarily robust either across similar populations or even within the same population, especially the very small clusters/subgroups. These subgroups would be more meaningful/convincing if the authors could show at the least that they are stable or they associate meaningfully to later health outcomes such as damage progression (through use of X-rays) or functional disability."

2.Related to my Comment 4 in my previous review, I believe it is unsatisfactory to not provide the cross-tabulations of the clustering structures of DAS28-ESR by CDAI for early RA and established RA, irrespective of whether some of the cells have small numbers. These cross-tabulations without much effort can be provided as supplementary material. Additionally the authors have not discussed what the implications of these differing clustering structures on treat-to-target strategies which aim to improve health outcomes of patients with RA.

3. In the Discussion, the authors added the following sentence "There is also a possibility of more clinic visits by patients with high disease activity compared to those with LDA status, which may affect the impression of disease course toward persistent high disease activity in these patients." This does not make sense to me as all patients analysed had the same number of visits (i.e. 5). The authors should clarify.

4. In the Data Analysis section, the authors need to add to the sentence "Subjects are then assigned to the group they most likely belong", the criterion (e.g. based on that group being estimated to have the highest posterior probability of the subject being allocated to it) that is used to make this allocation.

5. Table A1 Suppl: The column headings (Group1, Group2, Group3, Group4) do not make sense.

6. Table A2 Suppl: Based on the 24-month mean value of 12.8 for Group 6 (VHD-LDA), it is unclear why this group is described as VHD-LDA, when LDA based on CDAI is defined to have a CDAI value <= 10.

Reviewer #3: Thank you for revising this interesting manuscript according to our suggestions. I have no further comments.

7. PLOS authors have the option to publish the peer review history of their article (what does this mean?). If published, this will include your full peer review and any attached files.

Reviewer #1: No

Reviewer #2: No

Reviewer #3: No

---

## [Author Response · Author response to Decision Letter 1]

9 Aug 2022

6. Review Comments to the Author

Reviewer #1: (No Response)

Reviewer #2: 1. The authors have not addressed (or attempted to address) my concern regarding validation (Comment 3). As mentioned in my earlier review, "My concern is that the clusters/subgroups found are not necessarily robust either across similar populations or even within the same population, especially the very small clusters/subgroups. These subgroups would be more meaningful/convincing if the authors could show at the least that they are stable or they associate meaningfully to later health outcomes such as damage progression (through use of X-rays) or functional disability."

Authors’ response: Thanks for this comment. As recommended, we addressed stability by performing cross-tabulation of CDAI and DAS28 ESR in both early and established RA and added results in the supplementary material (Tables A3 and A6 Suppl). As an exploratory analysis, to show some meaningful association between DAS28 subgroups and functional disability (HAQ-DI and pain) we have added a table for early RA patients, in the supplementary material (Table A7 Suppl). We also added a related statement in the discussion.

2.Related to my Comment 4 in my previous review, I believe it is unsatisfactory to not provide the cross-tabulations of the clustering structures of DAS28-ESR by CDAI for early RA and established RA, irrespective of whether some of the cells have small numbers. These cross-tabulations without much effort can be provided as supplementary material. Additionally the authors have not discussed what the implications of these differing clustering structures on treat-to-target strategies which aim to improve health outcomes of patients with RA.

Authors’ response: Thanks for this comment. We did cross-tabulation of CDAI and DAS28 ESR in both early and established RA and added them in the supplementary material (Tables A3 and A6 Suppl). We also discussed the association between subgroups identified by CDAI and DAS28 and implication of CDAI and DAS28 as two most common disease activity indices in the clinic setting. We have added a paragraph on treat-to target strategies in the discussion.

3. In the Discussion, the authors added the following sentence "There is also a possibility of more clinic visits by patients with high disease activity compared to those with LDA status, which may affect the impression of disease course toward persistent high disease activity in these patients." This does not make sense to me as all patients analysed had the same number of visits (i.e. 5). The authors should clarify.

Authors’ response: Thanks for this comment. To clarify this, all patients had data at baseline, 6 month, 12 months, and 24 months follow-up. However, they may have had different numbers of visits over 24 months (Mean: 5.7; SD: 2.0). 

4. In the Data Analysis section, the authors need to add to the sentence "Subjects are then assigned to the group they most likely belong", the criterion (e.g. based on that group being estimated to have the highest posterior probability of the subject being allocated to it) that is used to make this allocation.

Authors’ response: Thanks for this comment. We added this to the sentence.

5. Table A1 Suppl: The column headings (Group1, Group2, Group3, Group4) do not make sense.

Authors’ response: Thanks for this comment. We removed them.

6. Table A2 Suppl: Based on the 24-month mean value of 12.8 for Group 6 (VHD-LDA), it is unclear why this group is described as VHD-LDA, when LDA based on CDAI is defined to have a CDAI value <= 10.

Authors’ response: Thanks for this comment. We corrected them.

Reviewer #3: Thank you for revising this interesting manuscript according to our suggestions. I have no further comments.

---

## [Decision Letter · Decision Letter 2]

25 Aug 2022

Disease Activity Trajectories for Early and Established Rheumatoid Arthritis:  Real-World Data from a Rheumatoid Arthritis Cohort

PONE-D-22-02930R2

Dear Dr. Movahedi,

We’re pleased to inform you that your manuscript has been judged scientifically suitable for publication and will be formally accepted for publication once it meets all outstanding technical requirements.

Kind regards,

Masataka Kuwana, MD, PhD

Academic Editor

PLOS ONE

Additional Editor Comments (optional):

Reviewers' comments:

Reviewer's Responses to Questions

**Comments to the Author**

1. If the authors have adequately addressed your comments raised in a previous round of review and you feel that this manuscript is now acceptable for publication, you may indicate that here to bypass the “Comments to the Author” section, enter your conflict of interest statement in the “Confidential to Editor” section, and submit your "Accept" recommendation.

Reviewer #2: All comments have been addressed

2. Is the manuscript technically sound, and do the data support the conclusions?

Reviewer #2: Yes

3. Has the statistical analysis been performed appropriately and rigorously? 

Reviewer #2: Yes

4. Have the authors made all data underlying the findings in their manuscript fully available?

Reviewer #2: No

5. Is the manuscript presented in an intelligible fashion and written in standard English?

Reviewer #2: Yes

6. Review Comments to the Author

Reviewer #2: The authors have addressed my comments now.

There are a few minor edits for the authors to make:

1. Disease Trajectories in early RA, CDAI section, p7: Please replace the labelling and description of Group 6 to VHD-MD from VHD-LDA and from low disease activity to moderate disease activity.

2. Disease Trajectories in early RA, CDAI section, last paragraph, p7: Although the sentence "Almost 60% of patients who were classified as MD-REM by CDAI were assigned to the LDA-REM group using DAS28, confirming disease remission at 24 months" is correct. In fact, 100% of these patients were in either the LDA-REM or MD-REM groups using DAS28. Therefore a much higher proportion than the 60% actual were in confirmed remission at 24 months. Therefore you can make the case even stronger.

3. Disease Trajectories in established RA, DAS28-ESR section, pp7-8: Table A3 Suppl should be Table A4 Suppl.

4. Discussion section, p 11, last paragraph: It is not correct that the 267 patients in the RA-MAP consortium were "untreated RA patients". They were untreated at enrollment in the study, but thereafter were treated. Please delete "untreated" and replace appropriately.

7. PLOS authors have the option to publish the peer review history of their article (what does this mean?). If published, this will include your full peer review and any attached files.

Reviewer #2: No

---

## [Editor Report · Acceptance letter]

26 Aug 2022

PONE-D-22-02930R2 

Disease Activity Trajectories for Early and Established Rheumatoid Arthritis:  Real-World Data from a Rheumatoid Arthritis Cohort 

Dear Dr. Movahedi:

I'm pleased to inform you that your manuscript has been deemed suitable for publication in PLOS ONE. Congratulations! Your manuscript is now with our production department. 

Kind regards, 

on behalf of

Prof. Masataka Kuwana 

Academic Editor

PLOS ONE